# Thymosin Beta 4 Inhibits LPS and ATP-Induced Hepatic Stellate Cells via the Regulation of Multiple Signaling Pathways

**DOI:** 10.3390/ijms24043439

**Published:** 2023-02-08

**Authors:** Jihye Choi, Yunsang Cho, Hwal Choi, Sangmin Lee, Hyeju Han, Jeonghyeon Lee, Jungkee Kwon

**Affiliations:** Department of Laboratory Animal Medicine, College of Veterinary Medicine, Jeonbuk National University, Iksan 54596, Republic of Korea

**Keywords:** autophagy, NLRP3 inflammasome, Thymosin beta 4

## Abstract

Risk signals are characteristic of many common inflammatory diseases and can function to activate nucleotide-binding oligomerization (NLR) family pyrin domain-containing 3 (NLRP3), the innate immune signal receptor in cytoplasm. The NLRP3 inflammasome plays an important role in the development of liver fibrosis. Activated NLRP3 nucleates the assembly of inflammasomes, leading to the secretion of interleukin (IL)-1β and IL-18, the activation of caspase-1, and the initiation of the inflammatory process. Therefore, it is essential to inhibit the activation of the NLRP3 inflammasome, which plays a vital role in the immune response and in initiating inflammation. RAW 264.7 and LX-2 cells were primed with lipopolysaccharide (LPS) for 4 h and subsequently stimulated for 30 min with 5 mM of adenosine 5′-triphosphate (ATP) to activate the NLRP3 inflammasome. Thymosin beta 4 (Tβ4) was supplemented to RAW264.7 and LX-2 cells 30 min before ATP was added. As a result, we investigated the effects of Tβ4 on the NLRP3 inflammasome. Tβ4 prevented LPS-induced NLRP3 priming by inhibiting NF-kB and JNK/p38 MAPK expression and the LPS and ATP-induced production of reactive oxygen species. Moreover, Tβ4 induced autophagy by controlling autophagy markers (LC3A/B and p62) through the inhibition of the PI3K/AKT/mTOR pathway. LPS combined with ATP significantly increased thee protein expression of inflammatory mediators and NLRP3 inflammasome markers. These events were remarkably suppressed by Tβ4. In conclusion, Tβ4 attenuated NLRP3 inflammasomes by inhibiting NLRP3 inflammasome-related proteins (NLRP3, ASC, IL-1β, and caspase-1). Our results indicate that Tβ4 attenuated the NLRP3 inflammasome through multiple signaling pathway regulations in macrophage and hepatic stellate cells. Therefore, based on the above findings, it is hypothesized that Tβ4 could be a potential inflammatory therapeutic agent targeting the NLRP3 inflammasome in hepatic fibrosis regulation.

## 1. Introduction

The activation of hepatic stellate cells (HSC) is a characteristic of hepatic fibrosis that can lead to cirrhosis, liver failure, and liver cancer [1,2]. Furthermore, the activation of HSC can limit the chronic wound healing reaction, which can result in portal hypertension and related complications [2]. There are generally four stages of liver fibrosis: (1) initiation induced by primary damage to organs, (2) activation of effector cells, (3) extension of extracellular substrates, and (4) impaired extracellular dynamic deposition and absorption [3]. As the fibrosis stage progresses, mediators containing inflammatory cytokines such as tumor necrosis factor (TNF)-α, transforming growth factor (TGF)-β, interleukin (IL)-1β, IL-6, IL-18 and oxygen are activated [4]. The continuation of such activation results in hepatic fibrosis and cirrhosis [5].

Other types of cells in the liver create networks and are related to hepatic fibrosis regulation [6,7]. Recent research has shown that as hepatic fibrosis progresses, macrophages induce the expression of IL-1β/IL-18 [7,8,9]. Hepatocellular damage caused by liver injury leads to the release of pro-fibrotic factors from infiltrating inflammatory cells, especially macrophages, resulting in the activation of HSCs [10]. When liver damage occurs, macrophages activated TGF-β, platelet-derived growth factors, and inflammatory cytokines and chemokines produce circulating monocytes and other inflammatory mediators [10,11,12]. Therefore, when constructing experimental models related to hepatic fibrosis, it is necessary to analyze the importance of macrophages in perpetuating and sustaining the inflammatory stage. TNF-α and IL-1β activated by macrophage-induced signals contribute to hepatocellular death, promote inflammatory cell recruitment, activate HSCs, and accelerate hepatic fibrosis [13,14]. HSCs activated by macrophage signals induce the nucleotide-binding oligomerization domain-containing protein 3 (NLRP3) inflammasome, which plays a fatal role in acute hepatic injury inflammatory response [15]. Therefore, HSC activation-related research must be essential for mechanism studies that cause hepatic fibrosis [16].

NLRP3 inflammasome activation plays an important role in the development and progression of liver disease [17]. Inflammasomes are assemblies of large caspase-1 activation complexes that sense exogenous and endogenous risk signals through the NLR involved in regulating innate immune responses [18]. Among the inflammasomes, the most studied NLRP3 inflammasome interacts with procaspase-1 through the apoptosis-associated speck-like protein containing a CARD (ASC) in response to a risk signal, inducing caspase-1 activation. Activated caspase-1 promotes the secretion of the inflammatory cytokines IL-1β and IL-18 [19,20], which induce the inflammatory activation of the NLRP3 inflammasome. The activation of the NLRP3 inflammasome requires a two-step reaction. The first signal, “priming,” up-regulates NLRP3 and pro-IL-1β through the activation of the transcription factor NF-kB. The second signal, “activation,” stimulates the assembly and activation of the NLRP3 inflammasome by a strong stimulus such as ATP, mitochondrial dysfunction, and K^+^ efflux [21,22]. Importantly, the activation of the NLRP3 inflammasome is closely related to the progression of hepatic fibrosis. Moreover, the nonalcoholic fatty liver mouse models require NLRP3 inflammasome activation for hepatitis inflammation and fibrosis [23]. In vitro studies show that IL-1β and IL-18 regulate HSC activation and promote hepatic fibrosis [24]. Therefore, we hypothesized that the inhibition of the NLRP3 inflammasome could regulate HSC activation and prevent hepatic fibrosis from progressing.

Several factors regulate the activation of NLRP3, and autophagic flow disruption is an important mechanism for NLRP3 activation [15]. Autophagy is a critical metabolic mechanism that can promote cell survival after cell damage. Previous studies have shown the role of autophagy in the activation of NLRP3 inflammasomes. The inhibition of autophagy in macrophages elevated the production of IL-1β, whereas the activation of autophagy decreased the secretion of IL-1β by targeting ubiquitinated inflammasomes or IL-1β for lysosomal degradation [25,26]. In addition, autophagy induction inhibits NLRP3 inflammasome activation by eliminating damaged mitochondria and blocking the release of mitochondrial reactive oxygen species (ROS) [27,28,29]. Autophagy has shown that it can regulate inflammasome activation through various mechanisms, such as by removing endogenous inflammasome activators such as ROS-producing mitochondria and removing inflammasomes and cytokines to help suppress NLRP3 inflammasome activation [30]. Such autophagy promotes hepatic regeneration by preserving mitochondrial integrity in hepatocytes and protecting them from aging [31]. In macrophages, autophagy was found to be an anti-inflammatory pathway with protective anti-fibrotic effects from peripheral secretion interactions [32], and in HSCs the activation of autophagy was also a potential core pathway to prevent liver fibrosis [33,34].

Thymosin beta 4 (Tβ4) is a ubiquitous protein that promotes wound healing and plays an essential role in various functions related to tissue regeneration, anti-apoptosis, anti-inflammation, cell proliferation, and differentiation [35,36,37]. Recently, we proposed that Tβ4 can regulate autophagy activation in PrP (106-126)-treated HT22 cells [38]. In particular, when carbon tetrachloride (CCl4) was added to human HSCs (LX-2 cells) to induce damage to liver cells, the PI3K/Akt pathway and apoptosis of cells are blocked by adding Tβ4, which may have potential as an effective liver fibrosis treatment [39,40,41]. According to recent research, autophagy is known to be suppressed and extended by mTOR and regulated by the PI3K/Akt pathway [42]. However, the effect of Tβ4 in NLRP3 inflammasome activation and autophagy and how this relates to macrophages and HSCs were not investigated. In this study, we evaluated the effect of Tβ4 on NF-kB and MAPK in the lipopolysaccharide (LPS) and adenosine triphosphate (ATP)-induced NLRP3 priming phase in macrophages or HSCs. In addition, we evaluated whether Tβ4 could inhibit NLRP3 inflammasomes through antioxidant activity and autophagy induction.

## 2. Results

### 2.1. Inhibitory Effect of Tβ4 on Cellular Viability of RAW264.7 and LX-2 Cells

The cytotoxicity of Tβ4 in RAW264.7 and LX-2 cells were investigated by using an MTT assay. The cells were cultured with Tβ4 at a concentration of 0.1~2 μg/mL for 24 h. As shown in Figure 1, Tβ4 had no significant cytotoxic effects (*p* > 0.05) on RAW264.7 and LX-2 cells. Therefore, Tβ4 at 0.1, 0.4, and 1.6 μg/mL were selected for further study.

### 2.2. Tβ4 Suppresses ROS Prudction Induced by Activation Signals

We examined the effects of Tβ4 on the production of ROS. LPS and ATP significantly increased the production of ROS compared to the control (Figure 2A). Tβ4 significantly reduced the LPS and ATP-increased production of ROS dose-dependently. LPS and ATP accelerated superoxide formation, a characteristic of mitochondrial ROS [16]. Therefore, we measured superoxide dismutase (SOD) activity to investigate whether Tβ4 inhibits LPS and ATP-induced superoxide formation (Figure 2B). As a result, Tβ4 increased LPS and ATP-reduced SOD activity in a dose-dependent manner. These results show that Tβ4 enhances SOD activity and attenuates ROS production.

We examined the effect of Tβ4 on the production of ROS in the LX-2 cell line. Various reports found that ROS were critically involved in NLRP3 inflammasome activation [43]. LPS and ATP induced the production of ROS in LX-2 cells, whereas Tβ4 pretreatment effectively inhibited this process (Figure 3A). SOD is a free radical scavenger that can remove superoxide anion radicals to protect cells from damage [44]. It was demonstrated that LPS and ATP significantly decreased SOD activity and that pretreatment with Tβ4 increased SOD activity (Figure 3B). These findings provide evidence that Tβ4 exhibit antioxidant effects in LPS and ATP-induced cells.

### 2.3. Tβ4 Regulates the Priming Signal of p38 MAPK Signaling Pathway

The effect of Tβ4 on NLRP3 priming through the NF-κB and JNK/p38 MAPK pathways were analyzed by Western blotting (Figure 4). As a result, cells treated with LPS (1 μg/mL) alone increased the protein expression of NF-κB, JNK/p38 MAPK, and pro-IL-1β and confirmed the induction of inflammation in the NLRP3 priming phase. The expression of NF-κB and JNK/p38 MAPK proteins increased by LPS confirmed a concentration-dependent reduction after treatment with Tβ4. In addition, Tβ4 inhibited an increase in IL-1β expression by NLRP3 priming. These results suggest that Tβ4 attenuates LPS-induced NLRP3 priming by inhibiting NF-kB and JNK/p38 MAPK expression.

The effects of Tβ4 on NF-κB and JNK/p38 MAPK pathways were analyzed by western blotting in LX-2 cells. As shown in Figure 5, after stimulation with LPS (1 μg/mL), the LX-2 cells increased the protein expression level of NF-κB and the JNK/p38 MAPK pathway. However, Tβ4 inhibited these effects (Figure 5) in a dose-dependent manner. In addition, Tβ4 inhibited an increase in IL-1β expression by NLRP3 priming in LX-2 cells. The results suggested that Tβ4 could suppress NF-κB and the JNK/p38 MAPK signaling pathway in LX-2 cells.

### 2.4. Tβ4 Improves Autophagy Associated with NLRP3 Inflammasome Inhibition

Western blotting was used to analyze the effects of Tβ4 on the autophagy pathway in LPS (1 μg/mL) and ATP (5 mM)-treated RAW264.7 cells (Figure 6). We found that the levels of p-PI3K, p-AKT, and p-mTOR were increased by LPS and ATP, whereas levels of p-PI3K, p-AKT, and p-mTOR were decreased by Tβ4 in a concentration-dependent manner. Furthermore, LPS and ATP significantly inhibited the expression of LC3A/B, an autophagy marker, and increased p62, another autophagy marker. Tβ4 significantly reversed the expression of autophagy markers upregulated by LPS and ATP treatment in a dose-dependent manner. These results demonstrate that Tβ4 promotes autophagy through PI3K/AKT/mTOR pathway inhibition in LPS and ATP-treated RAW 264.7 cells.

The impairment of the p38 MAPK pathway activation is associated with some inflammatory responses and causes downregulation accompanied by the inactivation of autophagy [45]. To further explore the role of MAPK signaling in autophagy inactivation, we evaluated LX-2 cells exposed to LPS and ATP with or without Tβ4. Western blot analysis showed that the regulation of p-PI3K, p-mTOR, p-AKT, and p62 after the addition of LPS and ATP was significantly restored by Tβ4 (1.6 μg/mL level) compared to cells treated with either LPS or ATP alone (Figure 7). Furthermore, LPS and ATP significantly inhibited the expression of LC3A/B, but Tβ4 significantly reversed the expression of this marker. These results further confirm that Tβ4 promotes autophagy by inhibiting the PI3K/AKT/mTOR pathway in LPS and ATP-treated RAW264.7 and LX-2 cells.

### 2.5. Tβ4 Regulates the Activation Signal of the NLRP3 Inflammasome

The effect of Tβ4 on the NLRP3 inflammasome state generated by LPS and ATP processing was evaluated. As shown in Figure 8, LPS and ATP-treated cells significantly increased the expression of the NLRP3 inflammasome-related proteins NLRP3, ASC, IL-1β, IL-18, and TNF-α compared to the control. Tβ4 markedly decreased the expression of NLRP3, ASC, IL-1β, IL-18, and TNF-α in a dose-dependent manner compared to LPS and ATP. In these results, the data indicated that Tβ4 inhibited the activation of the NLRP3 inflammasome.

To investigate whether LPS and ATP increase NLRP3 inflammasome priming signals, LX-2 cells were treated with various concentrations (0.1, 0.4, and 1.6 μg/mL) of Tβ4, and the expression levels of NLRP3 inflammasome-related proteins were evaluated. As shown in Figure 9, LPS and ATP-treated cells significantly increased the expression of NLRP3, IL-1β, IL-18, and Caspase-1 compared to the control. Tβ4 regulated the expression of NLRP3, IL-18, and Caspase-1 in a dose-dependent manner compared to LPS and ATP, and expression of IL-1β was increased by the 0.4 μg/mL concentration compared to 0.1 μg/mL, but decreased by the 1.6 μg/mL concentration. These results demonstrate that Tβ4 can control the LPS and ATP-activated NLRP3 inflammasome in RAW264.7 and LX-2 cells.

## 3. Discussion

Chronic liver diseases such as liver fibrosis, cirrhosis, hepatitis, and liver cancer are health problems that emerge constantly [46]. A common hallmark of hepatic fibrosis is chronic inflammation in which macrophages release several inflammatory cytokines and mediators [47]. Such inflammatory mediators induce HSC activation, which causes the acceleration of hepatic fibrosis [13,14]. In this study, we evaluated whether macrophages activate HSC. A change in the NF-kB/p38 MAPK pathway was observed when the medium of RAW264.7 stimulated by LPS for 24 h was added to LX-2 cells (Figure 5). In this in vitro experiment, changes in the LX-2 cell phenotype were dependent on inflammatory mediators secreted from macrophages. Recent reports suggest that LPS-induced RAW264.7 cells contain several inflammatory mediators such as TNF-α, IL-1β, IL-6, CCL2, and CXCL1 [48], and we also observed the expression of the NF-κB/p38 MAPK pathway and release of pro-IL-1β (Figure 4). When LX-2 cells were cultured in LPS-induced RAW264.7 cell medium, the expression of the NF-κB/p38 MAPK pathway and pro-IL-1β were increased compared with the untreated control group. The application of the indirect cell-cell interaction model shows that the inflammatory medium released from the activated macrophage triggers an inflammatory response in HSC. Thus, activated HSCs affect the hepatic fibrosis process and suggest that they are related to NLRP3 inflammasome activation.

The activation of the NLRP3 inflammasome plays an important role in various liver diseases, including nonalcoholic fatty liver and liver fibrosis [49]. When liver fibrosis progresses, chronic inflammation generally causes macrophages to secrete various inflammatory cytokines, which directly activate HSCs and affects NLRP3 inflammasome induction [47,50,51]. As in the published research, when the medium of LPS-activated RAW264.7 cells was used to culture LX-2 cells, the inflammatory mediators found in LX-2 cells were similar to those in RAW264.7 cells (Figure 4 and Figure 5) [47]. Because activated macrophages are not species-specific, two different cell species can mimic the interaction between cells [47,52]. After investigating the activation of HSC through macrophages, we added LPS (1 μg/mL) and ATP (5 mM) to each cell species to induce the activation of the NLRP3 inflammasome. NF-κB is an important signal pathway for HSC activation upon ATP-induced cytosolic Ca^2+^ by purinergic signaling receptors containing P2Y or LPS-induced TLRP4 stimulation [53,54]. Similarly, we identified that the overexpression of IL-1β in RAW264.7 and LX-2 cells primed with LPS and ATP could confirm this activation. The expression of IL-1β increased by LPS and ATP demonstrated that both RAW264.7 (Figure 8C) and LX-2 (Figure 9C) cells were significantly suppressed by Tβ4. These results determined that HSC and macrophage interactions affect the mechanism of activation state specificity [55], and inflammatory mechanisms resulting from these interactions are thought to contribute to the promotion of hepatic fibrosis.

Although Tβ4 has various functional roles including cell proliferation, wound healing, and anti-inflammation, the effects of this peptide on the NLRP3 inflammasome in macrophage and HSCs remain unknown. Recent studies have been conducted to study the effect of Tβ4 on liver fibrosis [56], but analysis of its role in the macrophage-HSC connection associated with the NLRP3 inflammasome was insufficient. In the current study, we used RAW264.7 and LX-2 cell models to investigate the effects of Tβ4 on the LPS and ATP-induced NLRP3 inflammasome in vitro. Here, we demonstrated in RAW264.7 and LX-2 cells that Tβ4 depletion significantly promoted the growth of both cells by activating the NLRP3 inflammasome. We observed that Tβ4 pretreatment could suppress IL-1β protein expression in LX-2 cells cultured with LPS-activated conditioned medium from RAW264.7 cells. These results suggest that Tβ4 can contribute to the prevention of chronic inflammation when liver fibrosis progresses.

Autophagy, which maintains cell homeostasis under cell stress, is known to prevent excessive inflammation by inhibiting the activation of the NLRP3 inflammation [25,57]. The LC3A/B protein is considered a marker of autophagy, as it is related to the extent of autophagosome formation and maturation [58]. The p62 protein, also called sequestosome 1 (SQSTM1), binds directly to the LC3 and GABARAP family proteins, and the protein itself is degraded by autophagy. Since p62 is accumulated when autophagy is inhibited and a reduced level is observed when autophagy is induced, p62 is used as an autophagy marker [59,60]. Therefore, we investigated autophagy induction by measuring LC3A/B and p62 in RAW264.7 and LX-2 cells. Tβ4 significantly increased LC3A/B expression and decreased the p62 expression in both cells. Autophagy not only regulates hepatocyte function but is also known to affect non-parenchymal cells such as macrophages and HSCs [61]. The induction of autophagy is known to protect against hepatocyte damage and hepatic fibrosis by contrasting clearly with the profibrogenic effect of autophagy in HSCs [34]. Recent studies have shown that macrophages promote cell-cell contact and are independently self-predatory, and the activation of HSCs and their regulation depends on the secretion pathway [62]. These results indicate that Tβ4 induces autophagy activation. The PI3K/AKT/mTOR signaling molecule is closely associated with the inhibition of autophagy [63]. For these reasons, the phosphorylated levels of these factors were measured to determine whether this pathway was involved in Tβ4-induced autophagy. As expected, Tβ4 significantly decreased the phosphorylation levels of PI3K/AKT/mTOR, which were increased by LPS and ATP. These results show that Tβ4 induced autophagy through the regulation of the PI3K/AKT/mTOR signaling pathway (Figure 6 and Figure 7).

To demonstrate the relevance of Tβ4 in NLRP3 expression, which is the rate-limiting step for inflammasome activation [64], we investigated NLRP3 inflammasome-associated factors. Our results are shown in Figure 8 and Figure 9. The increased expression of factors related to inflammasome activation by LPS and ATP stimulation was significantly decreased in a Tβ4 dose-dependent manner. NLRP3 interacts with the adaptor protein ASC, which recruits pro-caspase-1 to form a large protein complex called the NLRP3 inflammasome. IL-1β and IL-18 develop into their mature forms as a result of the induced activation of caspase-1. Tβ4 significantly reduces the expression of NLPR3, caspase-1, and ASC. At the same time, the NLRP3 inflammasome activation products IL-1β, IL-18, and TNF-α are also decreased.

Recent studies have demonstrated the association between autophagy, inflammasome activation, and cytokine processing [65]. They have also shown that autophagy regulates the activation of the NLRP3 inflammasome and other inflammasomes through various mechanisms [66]. The Tβ4 results confirm the regulation of autophagy and NLRP3 inflammasome activation in LPS and APS induction. The regulation of the NLRP3 inflammasome and autophagy presented in this study provides the negative and quantitative feedback loops necessary to balance cell-required host-defensive inflammatory responses with excessive inflammatory prevention. In addition, the activation of caspase-1 can inhibit the induction of autophagy by increasing the inflammatory response required for pathogen removal. Furthermore, concentrations of IL-1β were confirmed in the supernatant. The increased secretion of IL-1β by macrophages was found to activate the NLRP3 inflammasome in HSCs (Figure 8G). In such conditions, Tβ4 may act as a potential therapeutic agent in macrophages and HSCs, and may also prevent the progression of hepatic fibrosis. The molecular mechanisms that are considered in this study may produce balanced inflammatory outputs that favor tissue homeostasis recovery and prevent excessive lateral damage that can lead to autoimmune and anti-inflammatory pathology [67]. Tβ4 is useful in regulating the activation of these mechanisms when inflammatory conditions are induced by LPS and ATP.

We identified the mechanism underlying the anti-inflammatory effect of Tβ4 in RAW264.7 and LX-2 cells in LPS and ATP-induced inflammatory conditions. Studies have shown that Tβ4 can suppress the activation of the PI3K/Akt/mTOR pathway, NLRP3 inflammasome, and LPS and ATP-induced inflammation in macrophages and HSCs (Figure 10). Tβ4 attenuated inflammation through ROS-induced oxidative stress, confirming the protective effect of antioxidants in fibroblasts [35], and attenuates inflammation through the NF-κB p65 signaling pathway [68]. Our results show that Tβ4 reduced protein levels in the PI3K/Akt/mTOR pathway and in the NLRP3 inflammasome by regulating autophagy in LPS and ATP-induced RAW264.7 and LX-2 cells. In addition, NLRP3 knockdown significantly reduced LPS and ATP-induced activation and IL-1β secretion in NLRP3 inflammasomes, suggesting that NLRP3 mediates LPS and ATP-induced inflammation. The above results indicate that Tβ4 can suppress inflammation by suppressing NLRP3 inflammasomes overall. The limitation of the present study is that RAW264.7 cells and LX-2 cells cannot represent the macrophages or HSC of NLRP3 inflammasome conditions in vivo. For this reason, the detailed molecular mechanism in animal experiment of how Tβ4 regulates the NLRP3 inflammasome still needs to be elucidated in future studies.

## 4. Materials and Methods

### 4.1. Chemicals and Reagents

Tβ4 was purchased from Tocris Bioscience (Bristol, UK). Primary antibodies against LC3A/B, SQSTM1/p62, mammalian target of rapamycin (mTOR), phosphorylated (p)-mTOR, protein kinase B (AKT), p-PI3K, PI3K, p-AKT, p-JNK, JNK, IL-1β, TNF-ɑ, p-p38, and p38 were purchased from Cell Signaling Technology (Beverly, MA, USA). NLRP3, p-IL18, IL-18, p-NF-kB, NF-kB, and β-actin were obtained from Abcam (Cambridge, UK). Capase-1 was purchased from Santa Cruz (CA, USA). ASC was purchased from Invitrogen (Middlesex County, MA, USA). Secondary antibodies (i.e., anti-rabbit, anti-goat, or anti-mouse IgG antibody conjugated with horseradish peroxidase) were obtained from Millipore (Temecula, CA, USA). In addition, LPS and adenosine 5′-triphosphate (ATP) were purchased from Sigma (St. Louis, MO, USA). All other chemicals and reagents were of analytical grade.

### 4.2. Cell Culture and Treatment

Murine macrophage RAW264.7 cells (ATCC, Manassas, VA, USA) and human hepatic stellate LX-2 cells (ATCC, Manassas, VA, USA) are essential effectors of inflammation and the innate immune response. RAW264.7 cells and LX-2 cells were cultured in Dulbecco’s Modified Eagle’s Medium (DMEM, WelGene, Daegu, Korea) supplemented with 10% inactivation fetal bovine serum (FBS; WelGene), 100 U/mL of penicillin G, and 100 mg/mL of streptomycin (WelGene) at 37 °C under at 5% CO_2_ environment, and the medium was changed after 24 or 48 h.

RAW264.7 cells were cultured in 20 mL of the growth medium in 75 cm^2^ flasks with or without LPS (1 μg/mL) for 24 h. LX-2 cells were pre-cultured in 6-well plates for 24 h and then the culture medium was replaced with the conditioned medium obtained from RAW264.7 cells [47] for the NF-κB/p38 MAPK signaling pathway experiment. In addition, RAW264.7 cells and LX-2 cells were primed with LPS (1 μg/mL) for 4 h and subsequently stimulated for 30 min with 5 mM of ATP. Tβ4 (0.1, 0.4 and 1.6 μg/mL) was added to RAW264.7 and LX-2 cells 30 min before the addition of ATP for the autophagy and NLRP3 inflammasome inhibition experiments.

### 4.3. Measurement of Cell Viability

Cell viability was determined by MTT assay. RAW264.7 (1 × 10^4^ cells/well) and LX-2 (2 × 10^4^ cells/well) cells were seeded in 96-well plates containing 100 μL DMEM with 2% FBS. When the confluences reached 70~80%, the medium was changed, and dose concentrations of Tβ4 were added to the medium. After 24 h, 20 μL of 5 mg/mL MTT solution was added into wells and incubated for 4 h. After adding 150 μL of DMSO, the optical density was measured at 490 nm using a microplate reader (BioTek Instruments, Winooski, VT, USA).

### 4.4. Intracellular ROS Assay

Intracellular ROS concentration was measured using 2′7′-dichlorofluorescein diacetate (DCF-DA; Invitrogen). RAW 264.7 cells were seeded in a 48-well plate and cultured for 24 h. Cells were pretreated with Tβ4 (0.1, 0.4, and 1.6 µg/mL) for 1 h and then with LPS (1 μg/mL) for 4 h and ATP (5 mM) for 30 min. The cells were washed with PBS and then incubated with 10 μM DCF-DA in PBS for 30 min. ROS concentration was measured at 490 nm using a microplate reader (BioTek Instruments).

Intracellular ROS concentration in LX-2 cells was measured using a Cellular ROS Assay kit (ab186027, Abcam, UK) according to the manufacturer’s instructions. LX-2 cells were seeded in a 96-well plate and cultured for 24 h. Cells were pretreated with Tβ4 (0.1, 0.4, and 1.6 μg/mL) for 1 h and then with LPS (1 μg/mL) for 4 h and ATP (5 mM) for 30 min. The fluorescence was recorded using an excitation wavelength of 520 nm and an emission wavelength of 605 nm.

### 4.5. Superoxide Dismutase (SOD) Activity Assay

SOD activity was measured by using a SOD assay kit (706002, Cayman Chemical, Ann Arbor, MI, USA). SOD plays an essential role in the antioxidant defense mechanism as a catalyst for decomposing superoxide anions into oxygen molecules and hydrogen peroxide [15]. The assay was performed according to the manufacturer’s instructions. SOD activity was measured at 440-460 nm using a microplate reader (BioTek Instruments).

### 4.6. Preparation of Cellular Extracts and Western Blot Analysis

Total protein was extracted with a passive lysis buffer (E1941, Promega, Madison, WI, USA) containing a protease inhibitor cocktail (P3100, GenDEPOT, Katy, TX, USA) and a phosphatase inhibitor cocktail (P3200, GenDEPOT). The proteins (30 μg) were separated by 4-20% SDS-PAGE and transferred onto polyvinylidene fluoride (PVDF; #162177, Bio-Rad, Contra Costa, CA, USA) membranes. The blots were blocked with 5% bovine serum albumin (BSA; A0100-010, GenDEPOT, Katy, TX, USA) in PBS for 1 h at room temperature and incubated with specific antibodies at 4 °C overnight. The blots were incubated with an HRP-conjugated secondary antibody for 1 h at room temperature. Immuno-reactive bands were detected by a Super Signal West Dura Extended Duration Substrate (34076, Thermo, Middlesex County, MA, USA) following the manufacturer’s instructions. A Chemi Imager Analyzing System (Alpha Innotech, San Leandro, CA, USA) was used for densitometric analysis directly from the blotted membrane.

### 4.7. IL-1β Concentration Analysis in RAW264.7 Cells

The amount of IL-1β secreted from RAW264.7 cells was determined in media using a mouse IL-1β ELISA kit (MLB00C, R&D Systems, Minneapolis, MN, USA) according to the manufacturer’s instructions. The cell culture supernatants were acquired by centrifugation at 2000 rpm for 15 min to remove cell pellets, and the supernatants were stored at −80 °C until used. The IL-1β concentration was measured at 450 nm using a microplate reader (BioTek Instruments).

### 4.8. Statistical Analysis

All of the experiments in this study were performed at least three times. Moreover, all values were expressed as mean ± SEM. A statistical analysis was conducted using GraphPad Prism version 5.0. (GraphPad Software, San Diego, CA, USA). A Student’s *t*-test was used to determine statistical differences compared with the non-treated group (C) or the LPS and ATP-treated group; *p* < 0.05 was considered statistically significant.

## 5. Conclusions

This study demonstrated that Tβ4 could suppress inflammation induced by LPS and ATP by inhibiting the NLRP3 inflammasome and promoting its autophagy. Therefore, Tβ4 has potential clinical value in preventing and treating such inflammatory conditions.

## Figures and Tables

**Figure 1 ijms-24-03439-f001:**
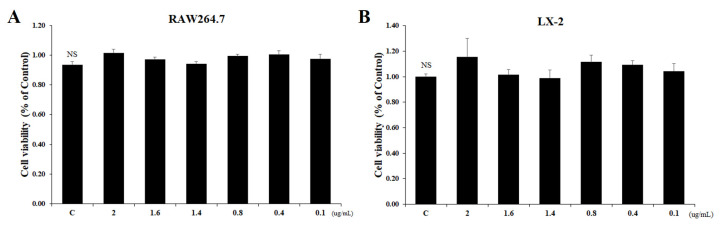
Effects of Tβ4 on cell viability measured by an MTT assay. (**A**) RAW264.7 cells; (**B**) LX-2 cells. All data are presented as means ± SEM from triplicate independent experiments. ^NS^ Indicates not significantly different with C (control) (*p* > 0.05).

**Figure 2 ijms-24-03439-f002:**
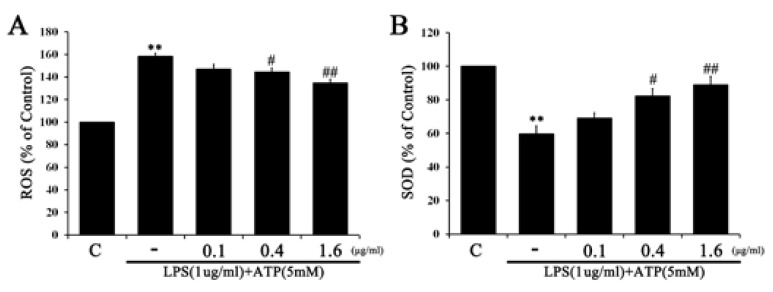
Effect of Tβ4 on ROS generation and SOD activity in LPS and ATP-induced RAW264.7 cells. RAW264.7 were pretreated with Tβ4 (0.1, 0.4, and 1.6 μg/mL) for 1 h, followed by NLRP3 inflammasome induction using LPS (1 μg/mL) for 4 h and ATP (5 mM) for 30 min. (**A**) ROS generation was evaluated by DCF-DA. (**B**) SOD expression was measured by ELISA in the cell lysates. Data are presented as mean ± SEM (*n* = 3). ** *p* < 0.01 compared with the C. ^#^
*p* < 0.05 compared with the LPS and ATP-treated group. ^##^
*p* < 0.01 compared with the LPS and ATP-treated group.

**Figure 3 ijms-24-03439-f003:**
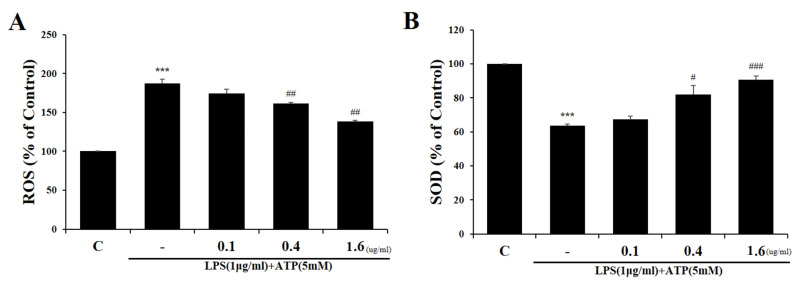
Effect of Tβ4 on ROS generation and SOD activity in LPS and ATP-induced LX-2 cells. LX-2 cells were pretreated with Tβ4 (0.1, 0.4, and 1.6 μg/mL) for 1 h, followed by NLRP3 inflammasome induction using LPS (1 μg/mL) for 4 h and ATP (5 mM) for 30 min. (**A**) ROS generation was evaluated by an ROS assay kit. (**B**) SOD expression was measured by ELISA in the cell lysates. Data are presented as mean ± SEM (*n* = 3). *** *p* < 0.005 compared with the C. ^#^
*p* < 0.05 compared with the LPS and ATP-treated group. ^##^
*p* < 0.01 compared with the LPS and ATP-treated group. ^###^
*p* < 0.001 compared with the LPS and ATP-treated group.

**Figure 4 ijms-24-03439-f004:**
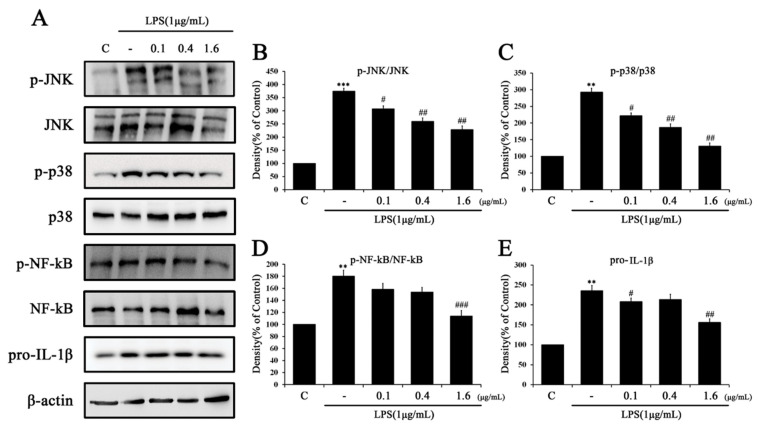
Effects of Tβ4 on the NF-κB pathway in LPS-induced RAW 264.7 cells. RAW 264.7 cells were pretreated with Tβ4 (0.1, 0.4, and 1.6 μg/mL) for 1 h, followed by NLRP3 priming using LPS (1 μg/mL) for 4 h. (**A**) The expression of (p)-JNK, (p)-p38, (p)-NF-kB, and pro-IL-1β was assessed by western blotting. (**B**–**E**) The bar graph represents the mean of A. The intensity of the control band was set to 100%, and the relative intensities of all other bands were calculated. Data are presented as mean ± SEM (*n* = 3). ** *p* < 0.01 compared with the control. *** *p* < 0.001 compared with the control. ^#^
*p* < 0.05 compared with the LPS-treated group. ^##^
*p* < 0.01 compared with the LPS-treated group. ^###^
*p* < 0.001 compared with the LPS-treated group.

**Figure 5 ijms-24-03439-f005:**
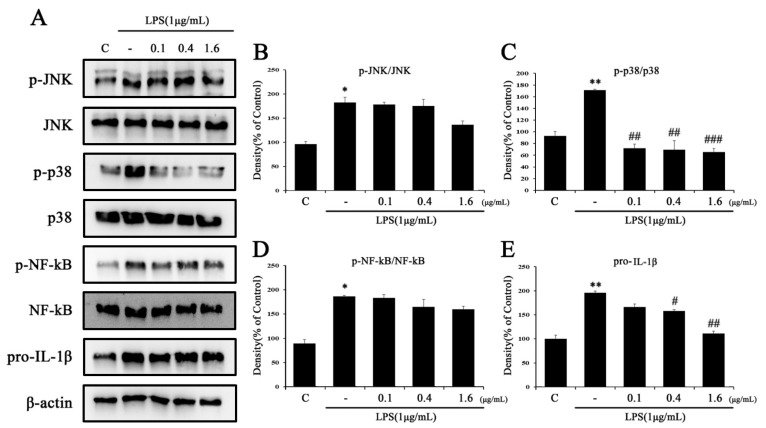
Effects of Tβ4 on the NF-κB pathway in LX-2 cells triggered by LPS-primed conditioned medium from RAW264.7 cells. LX-2 cells were pretreated with Tβ4 (0.1, 0.4, and 1.6 μg/mL) for 1 h, and LX-2 cells were supplemented with media from LPS-primed RAW264.7 cells for 24 h. (**A**) The expression of (p)-JNK, (p)-p38, (p)-NF-kB, and pro-IL-1β was assessed by western blotting. (**B**–**E**) The bar graph represents the mean of A. The intensity of the control band was set to 100%, and the relative intensities of all other bands were calculated. Data are presented as mean ± SEM (*n* = 3). * *p* < 0.05 compared with the C. ** *p* < 0.01 compared with the C. ^#^
*p* < 0.05 compared with the LPS-treated group. ^##^
*p* < 0.01 compared with the LPS-treated group. ^###^
*p* < 0.001 compared with the LPS-treated group.

**Figure 6 ijms-24-03439-f006:**
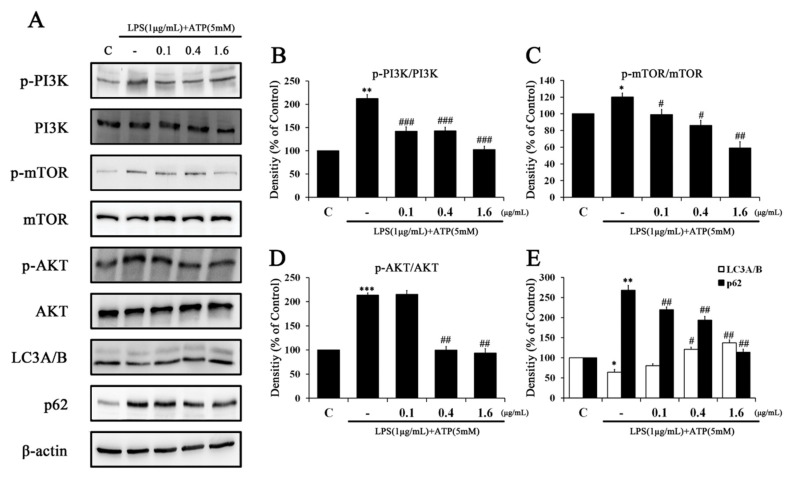
Effect of Tβ4 on the autophagy pathway in LPS and ATP-induced RAW264.7 cells. (**A**) The expression of (p)-PI3K, (p)-mTOR, (p)-AKT, LC3A/B, p62 and β-actin were assessed by western blotting. (**B**–**E**) The bar graphs indicate the average of A. The intensity of the control band was set to 100%, and the relative intensities of all other bands were calculated. Data are presented as mean ± SEM (*n* = 3). * *p* < 0.05 compared with the control. ** *p* < 0.01 compared with the control. *** *p* < 0.001 compared with the control. ^#^
*p* < 0.05 compared with the LPS and ATP-treated group. ^##^
*p* < 0.01 compared with the LPS and ATP-treated group. ^###^
*p* < 0.001 compared with the LPS and ATP-treated group.

**Figure 7 ijms-24-03439-f007:**
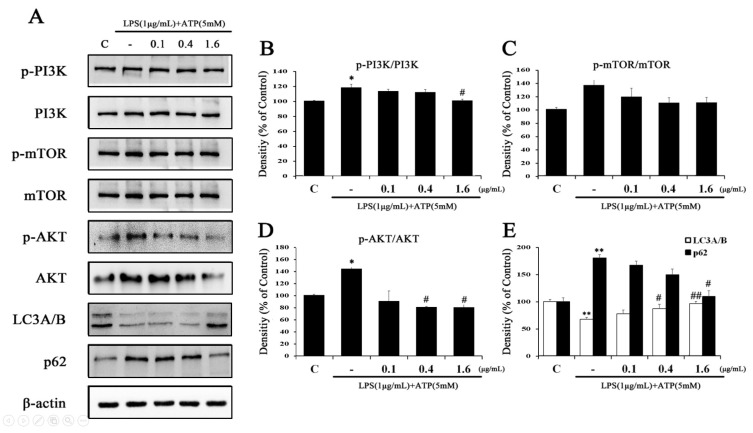
Effects of Tβ4 on the autophagy pathway in LPS and ATP-induced LX-2 cells. (**A**) The expression of (p)-PI3K, (p)-mTOR, (p)-AKT, LC3A/B, p62 and β-actin were assessed by western blotting. (**B**–**E**) The bar graphs indicate the average of A. The intensity of the control band was set to 100%, and the relative intensities of all other bands were calculated. Data are presented as mean ± SEM (*n* = 3). * *p* < 0.05 compared with the control. ** *p* < 0.01 compared with the control. ^#^
*p* < 0.05 compared with the LPS and ATP-treated group. ^##^
*p* < 0.01 compared with the LPS and ATP-treated group.

**Figure 8 ijms-24-03439-f008:**
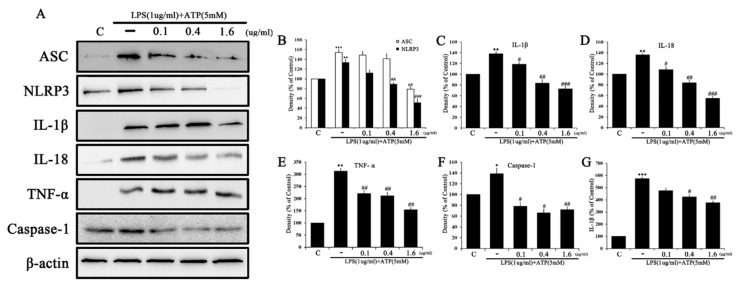
Effect of Tβ4 on NLRP3 inflammasomes in LPS and ATP-induced RAW 264.7 cells. (**A**) The expression of ASC, NLRP3, IL-1β, IL-18, TNF-α, Caspase-1 and β-actin were assessed by western blotting. (**B**–**F**) The bar graphs indicate the mean of A. The intensity of the control band was set to 100%, and the relative intensities of all other bands were calculated. (**G**) IL-1β was measured by ELISA in the cell supernatant. Data are presented as mean ± SEM (*n* = 3). * *p* < 0.05 compared with the control. ** *p* < 0.01 compared with the control. *** *p* < 0.001 compared with the control. ^#^
*p* < 0.05 compared with the LPS and ATP-treated group. ^##^
*p* < 0.01 compared with the LPS and ATP-treated group. ^###^
*p* < 0.001 compared with the LPS and ATP-treated group.

**Figure 9 ijms-24-03439-f009:**
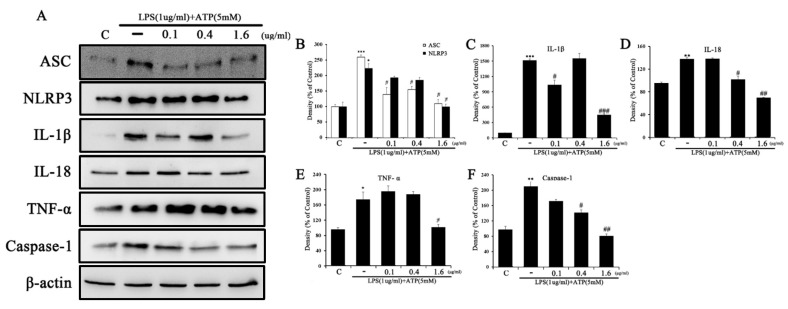
Effect of Tβ4 on NLRP3 inflammasomes in LPS and ATP-induced LX-2 cells. (**A**) The expression of ASC, NLRP3, IL-1β, IL-18, and Caspase-1 was assessed by western blotting. (**B**–**F**) The bar graphs indicate the mean of A. The intensity of the control band was set to 100%, and the relative intensities of all other bands were calculated. Data are presented as mean ± SEM (*n* = 3). * *p* < 0.05 compared with the control. ** *p* < 0.01 compared with the control. *** *p* < 0.001 compared with the control. ^#^
*p* < 0.05 compared with the LPS and ATP-treated group. ^##^
*p* < 0.01 compared with the LPS and ATP-treated group. ^###^
*p* < 0.001 compared with the LPS and ATP-treated group.

**Figure 10 ijms-24-03439-f010:**
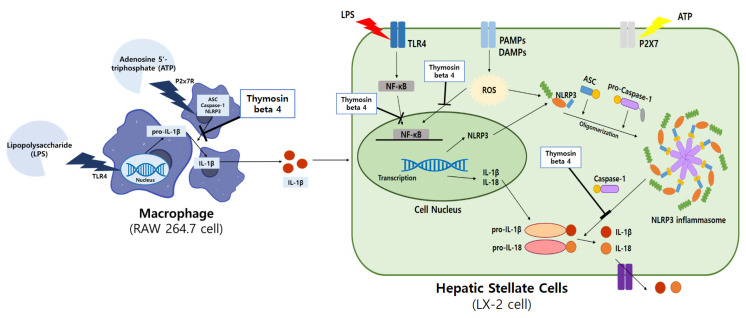
The role of Tβ4-mediated NLRP3 inflammasome activation in hepatic stellate cells activation through macrophage. LPS and ATP are involved in inflammatory crosstalk between hepatic stellate cell and macrophages.

## Data Availability

Not applicable.

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
