# Peer review of "Thymosin Beta 4 Inhibits LPS and ATP-Induced Hepatic Stellate Cells via the Regulation of Multiple Signaling Pathways"

_ijms, 2023, doi:10.3390/ijms24043439_

Round 1
Reviewer 1 Report
The manuscript entitled “Thymosin Beta 4 Inhibits LPS and ATP-Induced Hepatic Stellate Cells via the Regulation of Multiple Signaling Pathways” submitted by Choi et al., is an interesting work. The manuscript is well written. It is a solid work with clear hypothesis. The results presented are interesting and the discussion provided is good. This reviewer feels that the manuscript can be accepted for publication in the international journal of molecular sciences. Following are my suggestions that I would like to be incorporated for the benefit of research community.
1. Some relevant information about ATPs role should be provided in the introduction and that should be followed by its role in the current work.
2. Since the work reported has been performed on cell lines, I would like to see that the limitations of the reported work should be clearly indicated in the manuscript.
3. Some suggestions for future studies should be provided based on the expertise of the authors. These would be helpful in delineating the molecular mechanisms studies in future.
4. Page # 2 Line # 72 K+ should be K+.
Author Response
The manuscript entitled “Thymosin Beta 4 Inhibits LPS and ATP-Induced Hepatic Stellate Cells via the Regulation of Multiple Signaling Pathways” submitted by Choi et al., is an interesting work. The manuscript is well written. It is a solid work with clear hypothesis. The results presented are interesting and the discussion provided is good. This reviewer feels that the manuscript can be accepted for publication in the international journal of molecular sciences. Following are my suggestions that I would like to be incorporated for the benefit of research community.
On behalf of the authors, I would like to thank you for providing us the opportunity to improve our manuscript once again. We appreciated the careful reading of our manuscript as well as commenting and suggesting for better manuscript.
We have carefully rewritten and reorganized our manuscript according to the comments from you.
We hope that you agree with our manuscript that has been not only through revised but also strengthened by your comments.
Thank you for your kind consideration
- Some relevant information about ATPs role should be provided in the introduction and that should be followed by its role in current work
Response: ATP used in this study is known to induce a sustained increase in intracellular calcium inflow and activate NLRP3 inflammation to produce mature IL-1β. Although LPS can also cause the secretion of mature IL-1β, the secretion of IL-1beta induced by LPS can be further accelerated by ATP treatment in HSC. Therefore, in this study, IL-1β production was induced in LPS and ATP was treated to activate NLRP3 inflammation. In addition, the introduction in the manuscript does not include this information because it follows the ATP role identified in several studies. Thanks for your kind consideration for our manuscript.
- Since the work reported has been performed on cell line, I would like to see that the limitations of the reported work should be clearly indicated in the manuscript.
Response: Page #11, Line 375-376, we added the sentence “The limitation of the present study is that RAW264.7 cells and LX-2 cells cannot represent the macrophages or HSC of NLRP3 inflammasome condition at in vivo.” Following reviewer comment.
- Some suggestions for future studies should be provided based on the expertise of the authors. These would be helpful in delineating the molecular mechanisms studies in future.
Response: Page #11, Line 376-377, we added sentence “For this reason, the detailed molecular mechanism in in vivo of how Tβ4 regulating of NLRP3 inflammasome still need to be deeply elucidated in our future study.” Following reviewer comment.
- Page # 2 Line # 72 K+ should be K+.
Response: We changed the point following reviewer comment. Thanks for pointing out what we missed.

Reviewer 2 Report
This paper by Ji-Hye Choi et al describes that the peptide Tb4 can attenuate the NLRP3 inflammasome activation through multiple signaling pathways including autophagy in two different cell lines including macrophage and hepatic stellate cells (HSC). The authors showed that the Tb4 in macrophage cell line RAW264.7 and in hepatic cell line LX-2 1) by itself is not cytotoxic at various concentrations tested in two different cell lines 2) can enhance the SOD activity and subsequently reduce the ROS production 3) attenuates the NF-Kb signaling pathway in RAW264.7 cells and in LX-2 cells primed by the conditioned medium from RAW264.7 cells 4) activates the autophagy pathway by inhibiting PI3K/AKT/mTOR signaling 5) reduces the expression of NLRP3 inflammasome related proteins in both cell lines.
The manuscript is complete, written well, and has experiments to support the hypothesis.
Author Response
On behalf of the authors, I would like to thank you for providing us the opportunity to improve our manuscript once again.
Thank you for your kind consideration.